# Functional improvement is a better predictor of steady work than medical improvement for individuals with mental health conditions

Joshua C. Chang[1]*, Julia Porcino[1]*, Elizabeth Marfeo[1,2], Larry Tang[1,3], Howard Goldman[1,4], Elizabeth Rasch[1]

1 Rehabilitation Medicine Department, National Institutes of Health Clinical Center, Bethesda, Maryland, United States of America, 2 Department of Community Health, Tufts University, Medford, Massachusetts, United States of America, 3 Department of Statistics and Data Science, National Center for Forensic Science, University of Central Florida, Orlando, Florida, United States of America, 4 Department of Psychiatry, University of Maryland, School of Medicine, Baltimore, Maryland, United States of America

* josh.chang@nih.gov (JCC); julia.porcino@nih.gov (JP)

## Abstract

The Supported Employment Demonstration (SED) offered vocational and mental health services to recently denied disability benefit applicants with mental health conditions, along with other multiple co-morbidities, to evaluate the impact of evidence-based interventions on fostering employment and downstream benefits such as self-sufficiency, improved quality of life, and improved mental health. Using the SED public use file, we analyzed work outcomes for the study participants in relation to functional improvement, as measured by the Work Disability Functional Assessment Battery, vs. medical improvement. Using both Bayesian logistic regression models and neural networks, we found that functional improvement is a better predictor of steady work than medical improvement.

## Introduction

Employment and economic stability are critical social determinants of health and a key component of meeting sustainable development goals [1]. Labor force participation is known to have a positive effect on health, while conversely, loss of employment can have a negative impact as observed during the COVID-19 pandemic [2–7]. Participation in employment is particularly advantageous for individuals with mental health and physical limitations, as it fosters a sense of purpose, supports economic independence, and enhances emotional well-being. People with disabilities tend to have poorer health outcomes as well as lower rates of employment than their non-disabled peers [8]. Developing policies and interventions to help people with disabilities gain or maintain employment is critical for meeting public health goals and promoting health and well-being. Current models of disability incorporate a broad biopsychosocial perspective on factors that drive health and participation in social roles such as work. There are two key components to be able to apply these models

**Data availability statement:** This study used the SSA Supported Employment Demonstration Public Use File available at https://www.ssa.gov/disabilityresearch/sed_puf.html.

**Funding:** This research was supported, in part, by the Intramural Research Program of the National Institutes of Health Clinical Center (JCC, JP, EM, LT, HG, ER) and the U.S. Social Security Administration. All authors received salary from the National Institutes of Health Clinical Center. The funders had no role in study design, data collection and analysis, decision to publish, or preparation of the manuscript. The views, information or content, and conclusions presented do not necessarily represent the official position or policy of, nor should any official endorsement be inferred on the part of, the Clinical Center, the National Institutes of Health, or the Department of Health and Human Services.

**Competing interests:** The authors have declared that no competing interests exist.

to support programs for individuals with disabilities. First, it is important to know which factors inform and predict labor force participation. Second, there must be appropriate measures for these factors incorporated into research, policy, and support programs. In this work, we focus on the role of function and whether a self-report measure of whole-person function contributes to the prediction of employment participation and outcomes.

### The Work Disability Functional Assessment Battery (WD-FAB)

The Work Disability Functional Assessment Battery (WD-FAB) was developed to address gaps in data collection around function [9,10].The WD-FAB is a self-reported assessment of functional abilities that relate to work that provides scores in 8 scales across two domains of mental and physical functioning: Communication & Cognition, Resilience & Sociability, Self-Regulation, Mood & Emotions, Basic Mobility, Upper Body Function, Fine Motor Function, and Community Mobility. The WD-FAB is based on item response theory (IRT) and administered using computer adaptive testing (CAT) technology to make the assessment efficient and tailored to the respondent. IRT-CAT-based assessments use advanced psychometric methods to enhance the precision and efficiency of measurement by dynamically selecting the most relevant questions for each individual based on their previous responses. The WD-FAB has undergone rigorous development and psychometric testing, and recent efforts are now focused on applications of the WD-FAB across use cases. Previous work has considered whether certain thresholds or profiles of functioning are indicative of a person's ability to work [11]. From a longitudinal perspective, we want to understand how changes in WD-FAB scores relate to an individual's ability to return to work and maintain steady work.

### The Supported Employment Demonstration (SED)

The SED, a randomized controlled trial, aimed to understand whether providing work, behavioral, and health supports for recently denied Social Security disability benefit applicants can help such individuals achieve self-sufficiency. The study recruited individuals aged 18–49 who had a mental health impairment, alongside other co-morbidities, who were assigned to one of three study arms (full service, basic service, and usual service). Over the three-year study period, the study tracked work, quality of life, and income outcomes along with functioning information (provided by the WD-FAB), health condition, and health care utilization information. Functioning information was collected via the WD-FAB on an annual basis. Our goal was to understand the predictive power of functional improvement versus medical improvement on work outcomes with a particular focus on steady work [12–19] since, historically, the role of functional improvement is rarely considered, especially in relationship to medical improvement.

## Materials and methods

### Ethics statement

The NIH Office of Human Subjects Research Protection (OHRP) determined that this work does not meet the definition of human subjects' research pursuant to 45 CFR

46 and OHRP guidance. The views, information or content, and conclusions presented do not necessarily represent the official position or policy of, nor should any official endorsement be inferred on the part of, the Clinical Center, the National Institutes of Health, or the Department of Health and Human Services.

## Data

This study used the SED Public Use File (SED-PUF), a research dataset provided by the US Social Security Administration (SSA) that contains information collected from 2,944 individuals who participated in the SED. The accompanying codebook [20] provides the full list of variables and basic statistics for each variable.

## Statistical analyses

In this section we provide a high-level overview of our performed analyses. Please refer to the Section S1 in S1 Text for more details, and Section S3 in S1 Text for source codes.

We developed Bayesian hierarchical logistic regression models [21] to predict the annual odds of steady work for each individual in the SED study, where steady work refers to employment of at least half-time, as defined in the SED datasets. Bayesian methods handle missing values of predictors by averaging over estimates of these values. They are also able to use sparsity to find parsimonious models that are easier to understand. These models used three classes of covariates: demographic (age, employment history, education, housing status, vehicle access), medical – consisting of outpatient/emergency/inpatient utilization as well as assessments like the Drug Alcohol Screen Test (DAST [22]), Alcohol Use Disorders Identification Test (AUDIT [23–25]), Colorado Symptoms Index (CSI [26]), and function (WD-FAB mental and physical scores).

In predicting whether an individual had steady work in a given study year, we used both the baseline value of a predictor and the change from baseline for each predictor for that study year. To facilitate comparisons of effect sizes, we scaled baseline values of predictors by subtracting the mean and dividing by the standard deviation within our models. We scaled differences by subtracting the mean difference for a variable and scaling by the standard deviation of differences. Since Community Mobility scale scores had extreme missingness, which itself is informational, we omitted the scale scores in our modeling and instead used indicators for the presence of these scale scores. As a result of scaling, we are easily able to identify the impactful variables in our models by ranking the magnitude of their corresponding regression coefficients.

The SED-PUF contained a substantial number of missing values, particularly for function measurements. Notably, Community Mobility - Driving and Public Transportation (transit) - scales are not administered to respondents who do not use those transportation modes. Largely, for the seven remaining functioning scales, we found that participants were missing either zero, seven, fourteen, or twenty-one measurements. With the goal of retaining as many study participants as possible in the analysis, we performed missing value imputation within our model by considering possible values for the missing entries and then averaging over these values according to their probabilities.

## Fitting submodels

One main objective was to quantify the relative impact of function in predicting the likelihood of work, controlling for demographic differences. To this end, we also fitted submodels where we used the following sets of predictors: demographic only, demographic + function (omitting medical), demographic + medical (omitting functional), and demographic + medical + function (full model).

## Beyond linear regression

To ensure that our results are not due to the limitations of linear regression, we also replicated the same analysis using two nonlinear modeling techniques to investigate the consistency of results across different models. First, we fit Bayesian

 

artificial neural networks to the same data with a hidden layer size of 12. Second, we fit a piecewise Bayesian generalized linear model based on an additive method [27,28] that captures nonlinear effects in model coefficients. The model fits a separate regression model to each local region of the data. The reason for selecting these two nonlinear methods is that they both are piecewise linear models. For this reason they are related-to but also extend the generalized linear model in relaxing the assumptions between relationships. The logistic regression assumes a linear relationship between predictors and the log-odds of the outcome. The piecewise logistic regression and neural networks do not assume linearity in this relationship and provide flexible frameworks capable of capturing complex, non-linear associations and higher-order interactions among covariates.

## Model evaluation

We adjudicated the models on predictive accuracy using Bayesian leave one out (LOO) cross validation (CV). LOO-CV is a method used to evaluate models by training them multiple times, with one data point left out from the dataset during each training iteration. We estimated the LOO prediction for each observation modeled [29,30]. Using these quantities we estimated leave-one-out cross-validated receiver operator characteristic (ROC) and precision-recall (PRC) curves [31] and computed the area under these curves as a basis of comparison.

We used the Python bayesianquilts [27,28] wrapper for Tensorflow-probability [32] to perform all statistical analyses.

## Results

### Descriptive statistics

Ignoring missingness, the annual means and standard deviations of functioning variables (along with their changes from baseline) are presented in Table 1. Corresponding statistics for medical variables are available in Table 2. The distributions of WD-FAB scores by year are presented in Fig 1. The distributions of change relative to baseline in these scores is

**Table 1. Statistics mean (standard deviation) for outcome and functioning variables in the SED dataset, and their changes relative to baseline, ignoring missing values.**

| Variable | MDC90 | Overall | Baseline | Year 1 | Year 2 | Year 3 |
|---|---|---|---|---|---|---|
| *Outcome* | | | | | | |
| **Steady Work** | N/A | 0.173 | | 0.155 | 0.192 | 0.173 |
| *Physical and mental function* | | | | | | |
| Fine Motor Function Change (from baseline) | 8.2 | 42.7 (6.2) -0.1 (5.5) | 43.0 (5.6) – | 42.4 (6.4) -0.5 (5.6) | 42.7 (6.5) 0.4 (5.5) | 42.7 (6.5) -0.0 (5.2) |
| Upper Body Function Change | 4.6 | 39.3 (6.4) 0.1 (4.8) | 39.4 (5.8) – | 39.0 (6.5) -0.2 (4.8) | 39.3 (6.7) 0.3 (5.0) | 39.4 (6.6) 0.2 (4.7) |
| Communication & Cognition Change | 7.1 | 42.6 (7.8) 0.6 (6.9) | 41.8 (6.6) – | 42.4 (7.8) 0.7 (6.7) | 43.0 (8.4) 0.7 (7.0) | 43.5 (8.7) 0.5 (7.1) |
| Basic Mobility Change | 4.7 | 40.2 (6.5) 0.2 (4.7) | 40.2 (6.0) – | 40.2 (6.7) 0.3 (4.8) | 40.2 (6.6) 0.1 (4.9) | 40.3 (6.7) 0.1 (4.5) |
| Resilience Change | 8.2 | 48.3 (11.1) 0.6 (10.2) | 47.6 (9.6) – | 47.6 (11.3) 0.2 (10.1) | 48.9 (12.0 1.2 (10.1) | 49.4 (12.1) 0.5 (10.3) |
| Interpersonal Interactions Change | 8.6 | 46.5 (12.1) 1.5 (11.6) | 44.4 (8.8) – | 46.5 (12.3) 2.1 (10.6) | 47.6 (13.4) 1.3 (12.2) | 48.6 (14.3) 1.0 (12.1) |
| Mood & Emotions Change | 10.6 | 42.0 (14.1) 2.2 (13.1) | 38.6 (12.0) – | 42.5 (14.3) 4.0 (12.9) | 43.6 (15.1) 1.3 (13.5) | 44.6 (14.7) 1.0 (12.7) |
| Has Community Mobility (Ride) score | N/A | 0.21 | 0.31 | 0.22 | 0.17 | 0.13 |
| Has Community Mobility (Drive) score | N/A | 0.40 | 0.51 | 0.36 | 0.38 | 0.36 |
| Has Wheelchair Score | N/A | 0.03 | 0.04 | 0.03 | 0.03 | 0.03 |

**Table 2.** Statistics (mean and standard deviation) for selected medical variables: overall, at Baseline, Years 1-3, and changes relative to baseline.

| Variable | Overall | Baseline | Year 1 | Year 2 | Year 3 |
|---|---|---|---|---|---|
| Colorado Symptom Index (CSI) | 22.1 (12.4) | 25.2 (11.2) | 22.2 (12.7 | 20.4 (12.7) | 19.2 (12.5) |
| Change | -2.1 (10.3) | – | -3.0 (10.7 | -2.0 (10.2) | -1.0 (9.7) |
| Body Mass Index (BMI) | 31.5 (8.7) | 31.1 (8.9) | 31.5 (8.6) | 31.6 (8.5) | 32.2 (8.8) |
| Change | 0.1 (3.4) | – | -0.2 (3.7) | 0.1 (3.2) | 0.3 (3.3) |
| Inpatient Hospital Admissions | 0.3 (0.8) | 0.5 (0.9) | 0.3 (0.8) | 0.2 (0.8) | 0.2 (0.6) |
| Change | -0.1 (0.9) | – | -0.2 (1.0) | -0.1 (0.9) | -0.1 (0.8) |
| Drug Abuse Screening Test (DAST) | 0.8 (1.8) | 1.1 (2.0) | 0.7 (1.6) | 0.6 (1.5) | 0.6 (1.6) |
| Change | -0.2 (1.7) | – | -0.5 (1.9) | -0.0 (1.6) | 0.0 (1.5) |
| AUDIT | 2.9 (5.1) | 3.5 (5.7) | 2.7 (5.8) | 2.5 (4.6) | 2.5 (4.6) |
| Change | -0.3 (4.5) | – | -0.7 (4.9) | -0.1 (4.5) | -0.1 (4.1) |
| Total Emergency room visits | 0.9 (1.6) | 1.2 (1.9) | 0.9 (1.6) | 0.7 (1.5) | 0.6 (1.3) |
| Change | -0.2 (1.8) | – | -0.3 (1.0) | -0.1 (0.9) | -0.1 (0.8) |
| Emergency room drug visits | 0.01 (0.2) | 0.03 (0.3) | 0.01 (0.1) | 0.005 (0.07) | 0.003 (0.06) |
| Change | -0.01 (0.2) | – | -0.02 (0.3) | -0.01 (0.1) | -0.002 (0.1) |
| Emergency room physical visits | 1.0 (1.6) | 1.0 (1.6) | 0.7 (1.4) | 0.6 (1.5) | 0.4 (1.1) |
| Change | -0.2 (1.5) | – | -0.2 (1.7) | -0.2 (1.5) | -0.1 (1.4) |
| Emergency room mental visits | 0.2 (0.8) | 0.2 (0.8) | 0.1 (0.5) | 0.1 (0.4) | 0.0 (0.3) |
| Change | -0.1 (0.7) | – | -0.1 (0.9) | -0.0 (0.6) | -0.0 (0.5) |
| Total inpatient nights | 1.9 (6.6) | 1.9 (6.6) | 1.0 (4.2) | 1.0 (4.7) | 0.9 (3.8) |
| Change | -0.1 (0.9) | – | -0.9 (7.4) | -0.0 (5.5) | -0.3 (5.3) |
| Total ER visits | 0.85 (1.6) | 1.2 (1.9) | 0.88 (1.6) | 0.71 (1.5) | 0.59 (1.3) |
| Change | -0.23 (1.8) | – | -0.29 (2.0) | -0.21 (1.7) | -0.18 (1.6) |
| Total ER physical visits | 0.67 (1.4) -0.19 (1.5) | 0.98 (1.6) – | 0.73 (1.4) -0.25 (1.7) | 0.56 (1.5) -0.17 (1.5) | 0.41 (1.1) -0.15 (1.4) |
| Total ER mental visits | 0.11 (0.55) -0.05 (0.66) | 0.21 (0.82) – | 0.12 (0.49) -0.09 (0.86) | 0.07 (0.44) -0.05 (0.58) | 0.05 (0.33) -0.02 (0.48) |
| Total ER drug visits | (0.15) -0.01 (0.19) | 0.03 (0.26) – | 0.01 (0.12) -0.02 (0.28) | 0.005 (0.07) -0.01 (0.13) | 0.003 (0.06) -0.002 (0.09) |
| Total ER alcohol visits | (0.14) -0.01 (0.17) | 0.020 (0.204) – | 0.012 (0.140) -0.008 (0.21) | 0.008 (0.126) -0.004 (0.15) | 0.004 (0.071) -0.004 (0.13) |
| Admitted after alcohol ER visit | 0.007 (0.1) | 0.012 (0.141) | 0.004 (0.071) | 0.007 (0.110) | 0.003 (0.052) |
| Change | -0.003 (0.13) | – | -0.007 (0.15) | 0.003 (0.118) | -0.005 (0.117) |
| Admitted after drug ER visit | 0.007 (0.11) | 0.014 (0.50) | 0.006 (0.09) | 0.004 (0.07) | 0.002 (0.04) |
| Change | -0.004 (0.14) | – | -0.008 (0.19) | -0.002 (0.11) | -0.003 (0.08) |
| Admitted after mental ER visit | 0.06 (0.36) | 0.12 (0.50) | 0.068 (0.36) | 0.038 (0.28) | 0.025 (0.24) |
| Change | -0.03 (0.45) | – | -0.05 (0.55) | -0.03 (0.41) | -0.01 (0.35) |
| Admitted after physical health ER | 0.16 (0.57) | 0.23 (0.66) | 0.16 (0.56) | 0.14 (0.56) | 0.11 (0.50) |
| Change | -0.04 (0.65) | – | -0.07 (0.70) | -0.02 (0.66) | -0.03 (0.60) |
| Admitted after other ER visit | 0.02 (0.17) | 0.02 (0.16) | 0.02 (0.19) | 0.02 (0.19) | 0.01 (0.13) |
| Change | -0.003 (0.24) | – | 0.004 (0.24) | -0.007 (0.26) | -0.005 (0.22) |
| ER visits for other problems | 0.07 (0.56) | 0.08 (0.94) | 0.09 (0.38) | 0.05 (0.39) | 0.05 (0.28) |
| Change | -0.01 (0.71) | – | 0.004 (1.0) | -0.03 (0.52) | -0.006 (0.47) |
| Hospital stays for drug problems | 0.003 (0.06) | 0.005 (0.08) | 0.003 (0.07) | 0.002 (0.04) | 0.002 (0.05) |
| Change | -0.001 (0.08) | – | -0.002 (0.10) | -0.001 (0.08) | 0.00 (0.06) |
| Hospital stays for mental health | 0.024 (0.18) | 0.04 (0.25) | 0.03 (0.19) | 0.02 (0.14) | 0.01 (0.12) |
| Change | -0.011 (0.24) | – | -0.02 (0.29) | -0.01 (0.23) | -0.004 (0.18) |

*(Continued)*

Table 2. (Continued)

| Variable | Overall | Baseline | Year 1 | Year 2 | Year 3 |
|---|---|---|---|---|---|
| Hospital stays for physical health | 0.058 (0.29) -0.02 (0.39) | 0.09 (0.38) – | 0.06 (0.28) -0.03 (0.46) | 0.05 (0.27) -0.009 (0.37) | 0.04 (0.21) -0.013 (0.33) |
| Hospital stays for other problems | 0.01 (0.11) | 0.02 (0.12) | 0.01 (0.12) | 0.01 (0.11) | 0.01 (0.10) |
| Change | -0.002 (0.16) | – | -0.002 (0.17) | -0.003 (0.16) | -0.001 (0.15) |
| Routine outpatient mental visits | 3.7 (7.9) 0.36 (8.3) | 2.2 (4.6) – | 5.5 (9.7) 3.4 (8.9) | 4.0 (8.5) -1.5 (8.2) | 3.3 (7.8) -0.7 (6.6) |
| Self-help group visits | 0.90 (4.5) 0.00 (4.6) | 0.66 (2.9) – | 1.3 (5.5) 0.67 (5.0) | 0.93 (4.8) -0.40 (4.8) | 0.66 (4.5) -0.27 (4.1) |
| Public clinic visits | 0.31 (1.5) 0.02 (2.1) | 0.21 (1.1) – | 0.48 (2.1) 0.28 (2.1) | 0.29 (1.2) -0.19 (2.3) | 0.26 (1.3) -0.03 (1.8) |
| Private outpatient physician visits | 1.3 (2.7) 0.16 (3.0) | 0.75 (1.8) – | 1.8 (3.2) 1.3 (3.1) | 1.4 (2.7) -0.37 (3.1) | 1.2 (2.7) -0.19 (2.7) |
| Outpatient psychiatric visits | 0.96 (2.1) 0.11 (2.6) | 0.55 (1.5) – | 1.4 (2.7) 0.88 (2.8) | 1.0 (2.1) -0.40 (2.8) | 0.87 (2.1) -0.15 (2.1) |
| Outpatient other mental health visits | 1.6 (4.1) 0.26 (4.2) | 0.82 (2.1) – | 2.3 (4.8) 1.5 (4.4) | 1.8 (4.6) -0.52 (4.2) | 1.6 (4.2) -0.22 (3.7) |
| Outpatient other professional visits | 0.24 (1.2) -0.02 (1.7) | 0.18 (1.2) – | 0.43 (1.7) 0.25 (2.1) | 0.21 (0.92) -0.22 (1.8) | 0.14 (0.71) -0.075 (0.89) |
| Other outpatient visits | 0.24 (1.2) 0.04 (1.7) | 0.13 (0.92) – | 0.30 (1.4) 0.17 (1.4) | 0.24 (1.6) -0.05 (1.9) | 0.24 (1.4) -0.02 (1.9) |

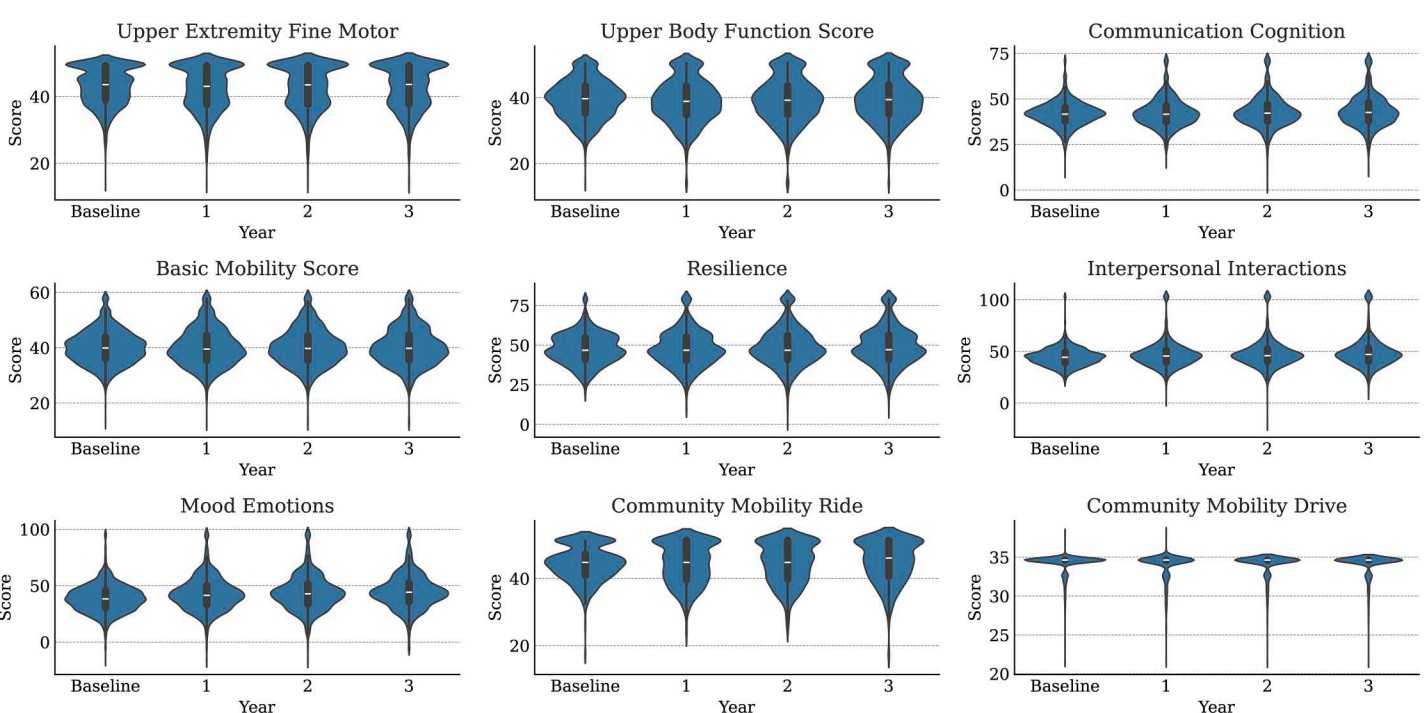

**Fig 1. Distributions of WD-FAB scores for each study year by scale.**

presented in Fig 2. As seen in Table 1, approximately 60% of Driving and 80% of Public Transportation Community Mobility scores were missing. As mentioned in the Methods, this fact motivated us to only incorporate their presence rather than the scores themselves into our predictive models. Except for Community Mobility, most study participants were missing either zero, seven, or fourteen functioning measurements (Fig 3).

We analyzed the impact of our missing value imputation marginalization scheme by repeating the analysis while retaining all study participants with at most zero, seven, and fourteen missing functioning measurements (excepting Community Mobility). We found that the differences between the models were minimal (Figure A in S1 Text). For this reason, we report only on the most-inclusive model (accepting a tolerance of 14 missing scale scores).

## Predictors of steady work

**Logistic regression analyses.** We standardized all predictor variables used in our models so that their effect sizes are directly comparable. In Fig 4, we display the odds ratios for the top 32 predictors for steady work, where the mean and 95% credible intervals are annotated. The predictors with the largest effects were: baseline working status (demographic), change in Communication & Cognition score (function), having a Community Mobility score (function), change in Upper Body Function score (function), "Other" race (demographic), having a bachelor's degree (demographic), having worked in the past 2 years at baseline (demographic), being in a treatment arm of the study (demographic), baseline Upper Body Function score (function), change in BMI (medical), and mental health-related ER visits (medical).

Fig 5 shows the top ten most impactful predictors of steady work when restricted to demographic variables, medical + demographic variables, and WD-FAB + demographic variables. The baseline working status was the top predictor in all three models. When looking at demographic variables only, the top predictors are related to work history and education. Additionally, being in a treatment arm of the study is predictive of achieving steady work.

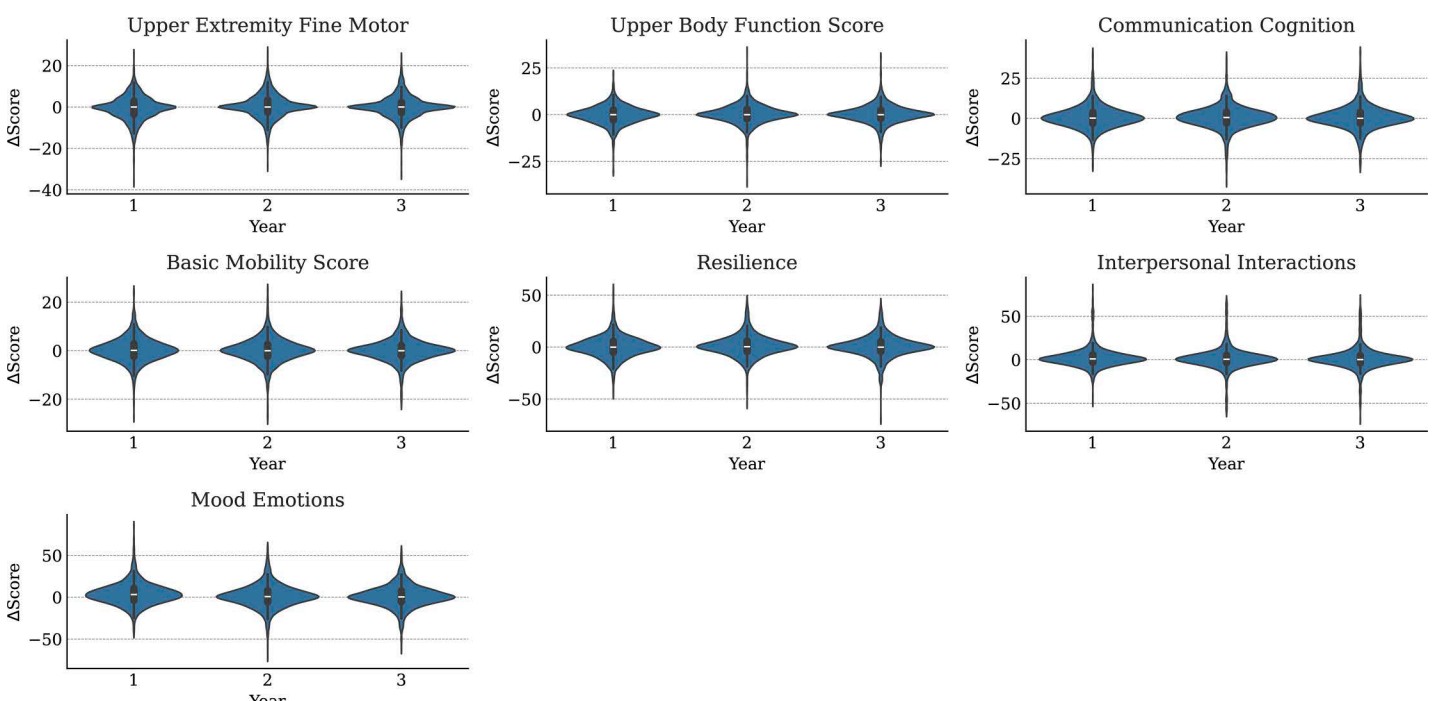

**Fig 2. Distributions of WD-FAB changes in function (relative to baseline) for each study year.** Community Mobility scale omitted.

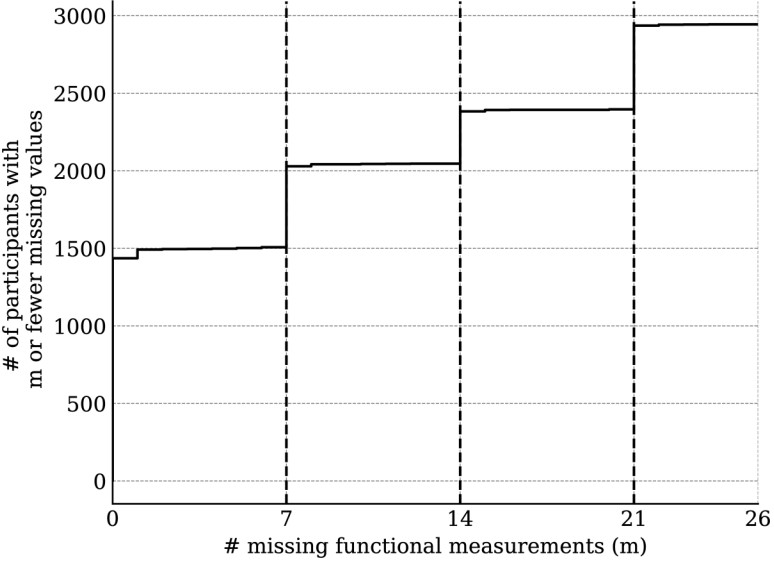

**Fig 3. Number of participants with at most m missing function measurements (not including Community Mobility scales).**

When adding WD-FAB measures, improvements in Communication and Cognition, Resilience, and Upper Body Function were the most influential predictors of steady work. Additionally, having a Community Mobility Drive score (implying that a person can operate a vehicle) is also a positive predictor of steady work. Ignoring the WD-FAB and adding medical predictors, both the BMI and the change in BMI are positively associated with steady work whereas drug related ER visits at baseline is negatively associated.

Fig 6 presents cross-validation-based model classification metrics, specifically the ROC and PR curves for each logistic regression model. Overall, these metrics provide an estimate of how well a given model can predict new outcomes based on new data.

**Nonlinear models.** In Fig 7, we present classification metrics for each of the non-linear model types that we fitted: Piecewise generalized linear regression, and Bayesian neural network. The classification performance of these two types was remarkably similar. Both types of models performed best when using the demographic + WD-FAB predictors, with that submodel performing better than the model fitted using all predictors.

## Discussion

### Functional information as a stronger predictor of steady work than medical and health care utilization data

In this manuscript, we leverage data from the SED to evaluate the relative predictive power of medical versus functional improvement in forecasting an individual's ability to maintain steady work on an annual basis, specifically among individuals with mental health conditions. While SSA has increasingly acknowledged the importance of functioning information in disability determinations, evaluation criteria still heavily rely on impairment data and healthcare utilization metrics as primary indicators of impairment severity. This approach often overlooks the direct impact of functional limitations on work capacity. Our findings strongly support that functional improvement, as measured by the WD-FAB, provides a more accurate prediction of work status for these individuals than traditional medical impairment and healthcare utilization measures.

As evident in Table 1, the average trend for change in function is slightly negative. However, large variability in these changes exists, indicating that a significant contingent of individuals shows improvement. The observed standard deviations of WD-FAB scale changes are comparable to their empirical test-retest minimal detectable change (MDC90)

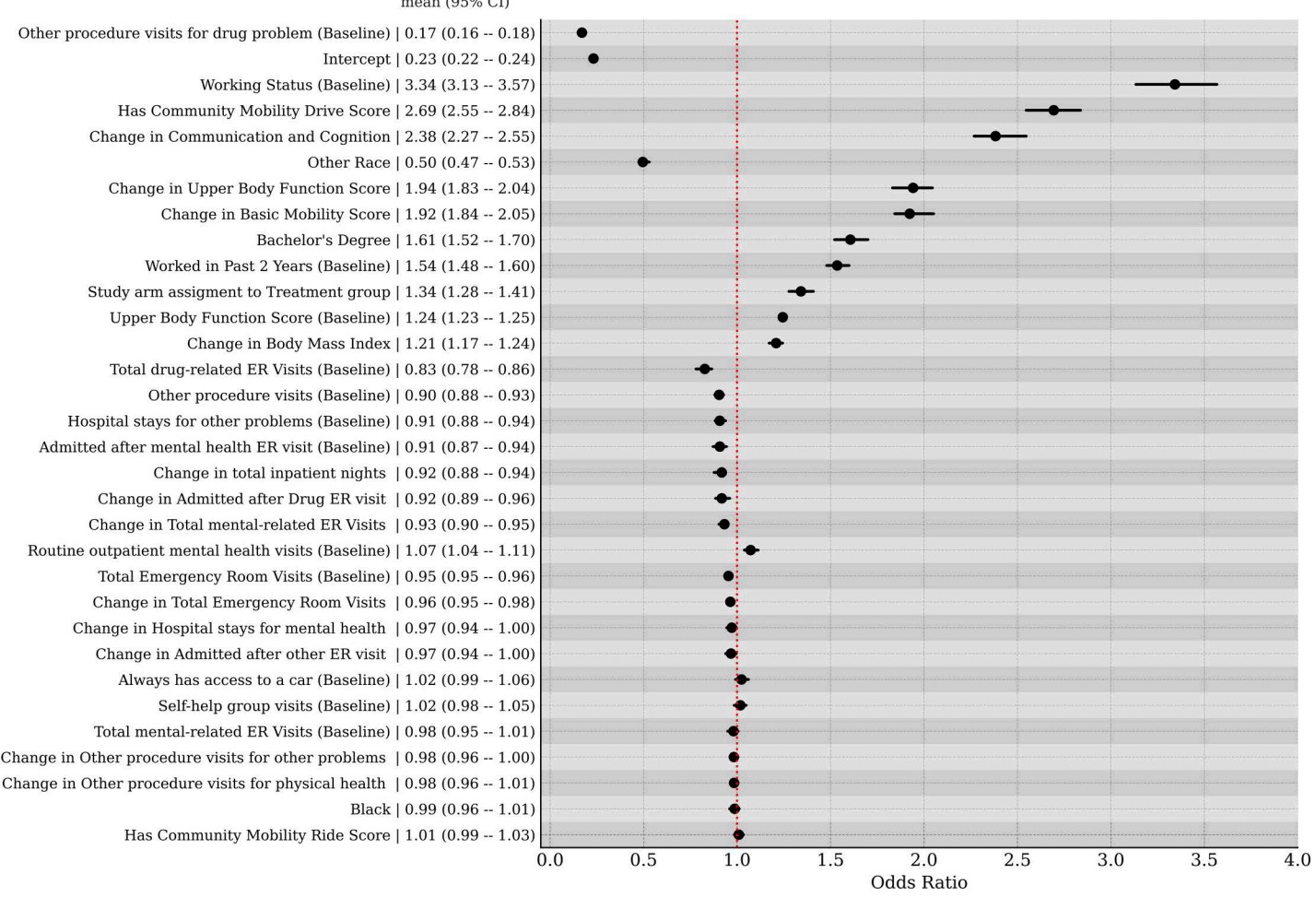

**Fig 4. Odds ratios for the top 32 predictors of steady work.** Units are standardized. Mean and 95% credible intervals presented.

thresholds [10], indicating that approximately one-third of participants experience functional improvements beyond the MDC90 threshold. However, even modest sub-threshold improvements portend increased odds of steady work. See thresholds presented in Table 1 for further details.

A detailed examination of the full predictive model (Fig 4) highlights that the top predictors of steady work are predominantly functional measures and their changes over time. Improvements in key functioning domains - such as Communication & Cognition, Upper Body Function, and Basic Mobility - emerge as strong predictors of steady work in each study year. Furthermore, when medical predictors are removed from the model (Fig 5), improvements in Resilience also become a significant factor in predicting sustained employment. Notably, while the participants were recruited into the SED based on their mental health conditions, a significant proportion also had co-occurring physical limitations. By considering a multidimensional profile of function encompassing both mental and physical domains, we obtain a more comprehensive and accurate measurement of overall ability and work potential.

Among the top ten predictors of steady work in the full model, change in BMI is the only medical variable, whereas functional measures dominate. Unlike work-related physical and mental function, the nature of the relationship between BMI and ability to work is unclear. For instance, for some job types, having a high BMI might be

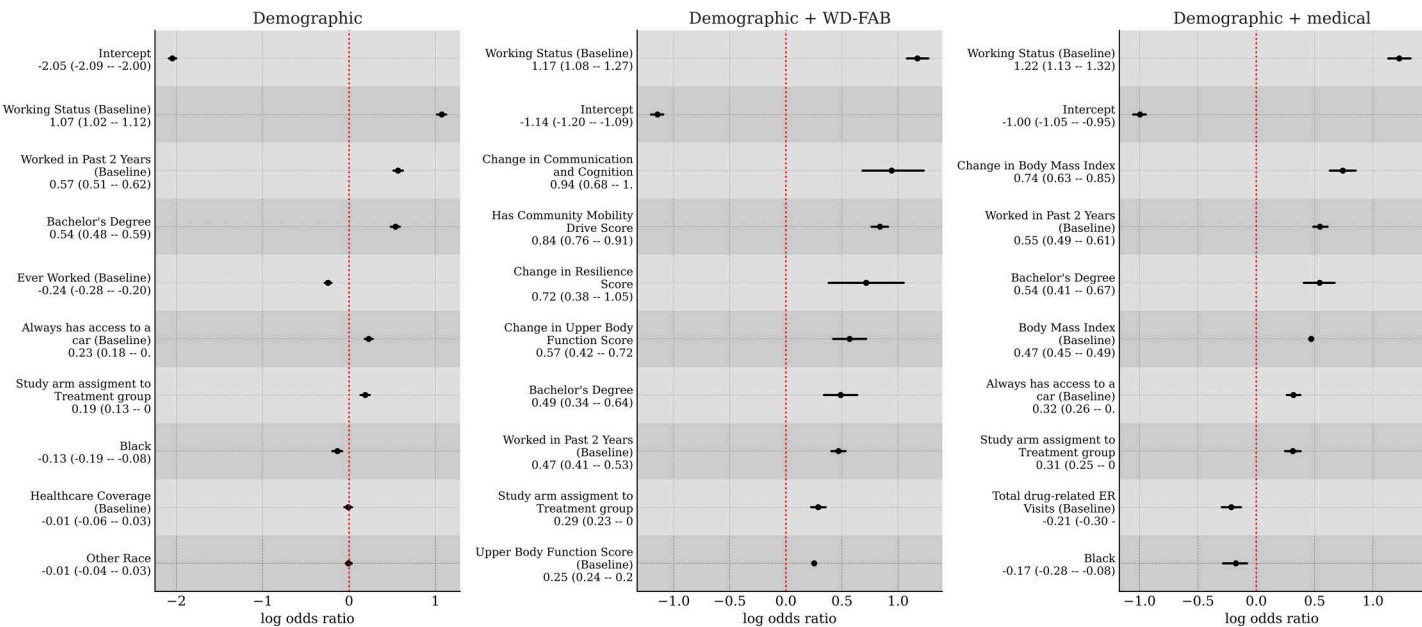

**Fig 5. Top predictors for steady work when restricted to demographic, demographic+function, or demographic+medical variables.**

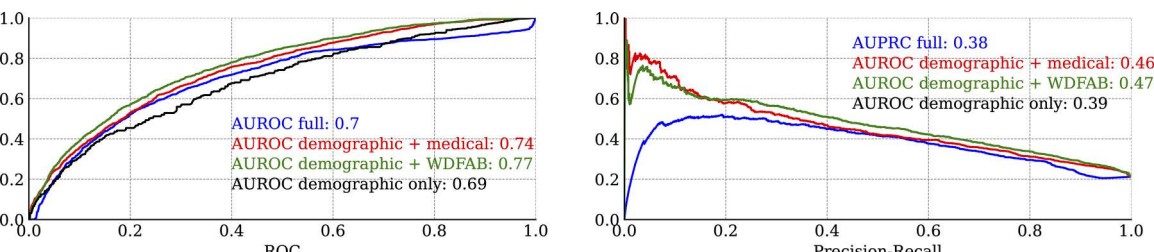

**Fig 6. Leave One Out (LOO) cross validated classification metrics for logistic regression models.** (a) Receiver operator characteristic and corresponding area under the curve. (b) Precision-recall curve.

an impediment to being able to work. While high BMI is generally associated with lower physical function [33], this relationship is not monotonic [34]. Certain debilitating physical conditions such as cancer or metabolic disorders may cause one to have a low BMI, where these disorders also impair physical/mental function in a way that makes work infeasible. In fact, there is evidence that having a higher BMI can be protective with respect to mortality [35,36]. Notably, when functional measures are removed (Fig 5), then the change in BMI becomes a much stronger predictor going from an average odds ratio of approximately 1.2 to approximately 2.1. This finding suggests that the increase (or decrease) of BMI is often associated to changes in the functional measures. Potentially, BMI and function are inter-related causal factors impacting the ability to work. For instance, functional decline could lead to weight gain which could then impact work ability. Some healthcare utilization variables, such as baseline mental health and substance-related ER visits, changes in total ER visits, and increases in inpatient nights, are negatively associated with steady work. Additionally, changes in overall ER visits and DAST scores appear as negative predictors of work ability when functional measures are removed from the model.

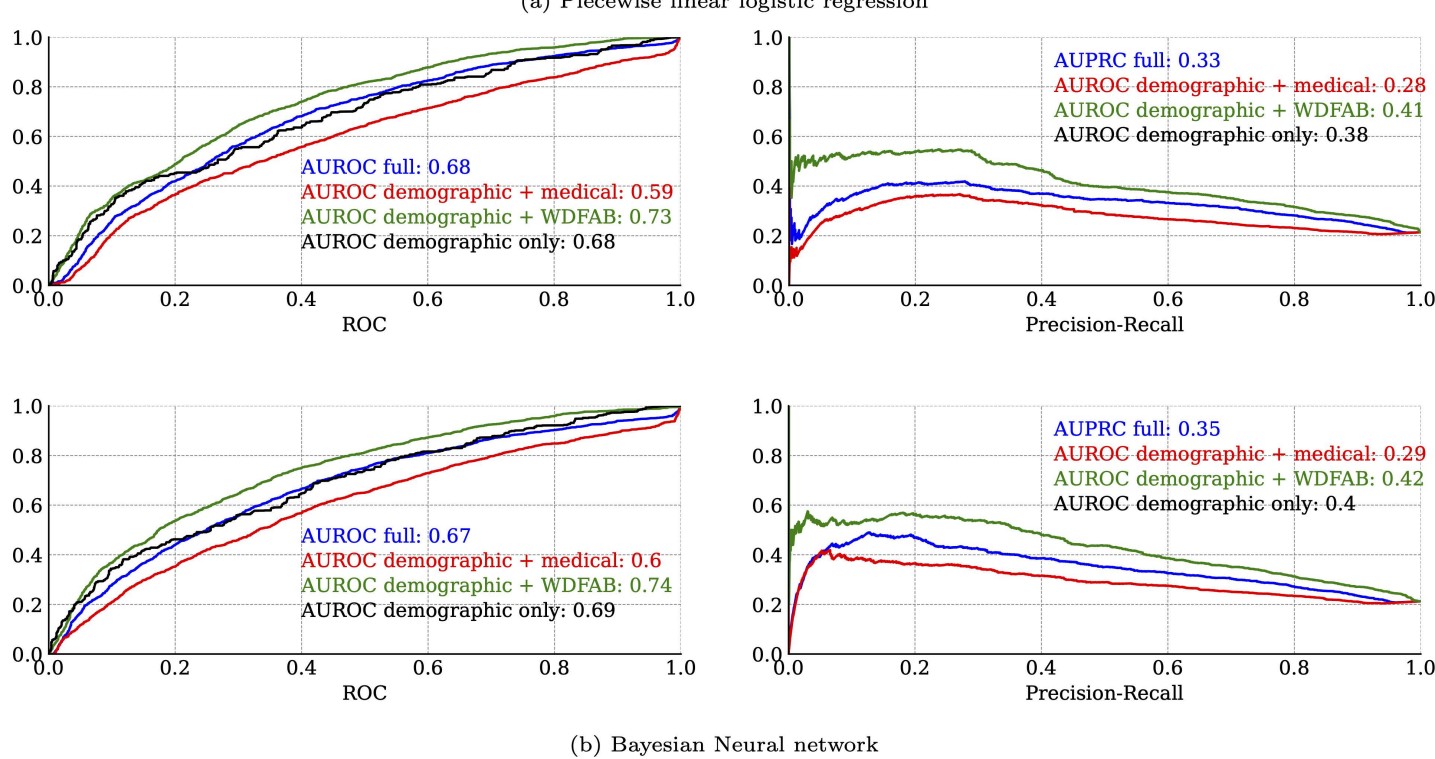

(a) Piecewise linear logistic regression

(b) Bayesian Neural network

**Fig 7. Leave One Out (LOO) cross validation classification metrics (ROC and PRC curves) for nonlinear models (a) Piecewise generalized linear (b) Bayesian neural network** AUROC refers to the area under the receiver operator curve, PRC refers to the area under the precision recall curve, larger values are better.

The WD-FAB characterizes how someone's physical and mental health limitations affect their ability to work along eight key subdomains of function: Basic Mobility, Upper Body Function, Fine Motor Function, Communication & Cognition, Resilience & Sociability, Self-Regulation, and Mood & Emotions [37]. The results imply that by adjusting for the subdomains of the WD-FAB, which reflect an overall functional profile, medical conditions are no longer strongly related to the outcome. These findings highlight how multidimensional functional assessments, such as the WD-FAB, can significantly outperform traditional indicators of disability - such as medical diagnoses and healthcare utilization - in predicting steady work outcomes.

Beyond the prominence of functional measures as top predictors of steady work, our analysis also demonstrates that models incorporating functional variables (i.e., the WD-FAB) alone have superior predictive accuracy compared to models relying solely on medical data. As illustrated in Fig 6, models that include demographic factors alongside WD-FAB scores achieve predictive accuracy better than the full model, as measured by both receiver operating characteristic (ROC) and precision-recall curves.

## Limitations and extensions

Our analysis is based on medical variables recorded in the SED Public Use File (SED-PUF). There may be additional medical variables, especially in the domain of condition-specific impairments that were not recorded in this dataset that are predictive of work outcomes. Additionally, we incorporated the medical variables directly whereas the function predictors are a low-dimensional representation of overall physical and mental function. It is possible that low-dimensional

representations of medical utilization [38] may be more predictive of work outcome than the original variables measured in this study.

## Conclusion

Functional improvement as measured by the WD-FAB is highly predictive of steady work, and more reliably predicts this outcome compared to medical impairment and healthcare utilization measures alone. These findings underscore the necessity of shifting disability assessment and work-capacity frameworks toward a more whole-person approach, moving beyond the reliance on medical diagnoses and healthcare utilization. By integrating multidimensional functional assessments such as the WD-FAB into processes and programs to help individuals with disabilities obtain and maintain employment, policymakers and practitioners can more accurately identify work potential and develop targeted interventions to support sustained employment for individuals with mental health conditions. In addition, determining one's ability to work is an important step in the disability determination process. The identified factors that significantly impact the ability to work will provide valuable insights in the determination.

## Supporting information

**S1 Text.  Supplemental methods and results.**
(PDF)

## Acknowledgments

We would like to acknowledge the contributions of Dr. Christine McDonough, who helped define the scope and interpretation of the study but passed away before the completion of this work.

## Author contributions

**Conceptualization:** Julia Porcino, Elizabeth Rasch.

**Data curation:** Julia Porcino.

**Formal analysis:** Joshua C. Chang.

**Funding acquisition:** Elizabeth Rasch.

**Investigation:** Julia Porcino.

**Methodology:** Joshua C. Chang.

**Project administration:** Julia Porcino.

**Software:** Joshua C. Chang.

**Supervision:** Julia Porcino, Elizabeth Rasch.

**Validation:** Joshua C. Chang.

**Visualization:** Joshua C. Chang.

**Writing – original draft:** Joshua C. Chang.

**Writing – review & editing:** Joshua C. Chang, Julia Porcino, Elizabeth Marfeo, Larry Tang, Howard Goldman, Elizabeth Rasch.

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
