## [Decision Letter · Decision Letter 0]

PMEN-D-25-00182

Functional improvement is a better predictor of steady work than medical improvement for individuals with mental health conditions

PLOS Mental Health

Dear Dr. Chang,

Thank you for submitting your manuscript to PLOS Mental Health. After careful consideration, we feel that it has merit but does not fully meet PLOS Mental Health’s publication criteria as it currently stands. Therefore, we invite you to submit a revised version of the manuscript that addresses the points raised during the review process.

We look forward to receiving your revised manuscript.

Kind regards,

David Onchonga, Ph.D.

Academic Editor

PLOS Mental Health

Additional Editor Comments (if provided):

Reviewers' comments:

Reviewer's Responses to Questions

**Comments to the Author**

1. Does this manuscript meet PLOS Mental Health’s publication criteria?

Reviewer #1: Yes

Reviewer #2: Yes

2. Has the statistical analysis been performed appropriately and rigorously?

Reviewer #1: Yes

Reviewer #2: Yes

3. Have the authors made all data underlying the findings in their manuscript fully available (please refer to the Data Availability Statement at the start of the manuscript PDF file)?

Reviewer #1: Yes

Reviewer #2: Yes

4. Is the manuscript presented in an intelligible fashion and written in standard English?

Reviewer #1: No

Reviewer #2: Yes

Reviewer #1: Thank you for the opportunity to review this important and well-executed study. Your manuscript, “Functional improvement is a better predictor of steady work than medical improvement for individuals with mental health conditions,” addresses a critical and underexplored topic in disability and vocational rehabilitation research. The use of the SSA-Supported Employment Demonstration (SED) dataset and the application of advanced statistical techniques, including Bayesian logistic regression, neural networks, and leave-one-out cross-validation make this a methodologically robust contribution. Your findings have direct relevance for public health and social policy, especially in the context of reforming disability evaluations toward a more function-focused approach.

That said, several areas of the manuscript would benefit from revision before it can be considered for publication. First, while the statistical methods and results are presented in a highly technical manner, the methods are so rigorous that they may not be easily accessible to the general readership of PLOS Mental Health. Consider simplifying explanations, summarizing key methodological decisions in the main text (e.g., the imputation strategy, feature scaling, rationale for variable selection), and referring to supplemental material only for technical details. Similarly, expanding the interpretation of the neural network results and their added value over logistic regression could enhance understanding.

Second, while the Discussion section is rich in implications, it could benefit from more critical reflection on why medical predictors underperformed. Is this due to measurement limitations, a conceptual mismatch, or perhaps the use of raw variables instead of derived composites? Clarifying this would strengthen the practical relevance of your findings. Additionally, results such as the positive predictive value of BMI warrant further exploration to avoid misinterpretation by readers. You may also wish to expand on how the WD-FAB might beented in real-world SSA evaluations or workplace settings. Finally, the manuscript is well written overall, but the clarity and narrative flow, feasibly implemparticularly in dense sections, would benefit from some restructuring and simplification. Figures should be explained more intuitively, and technical terms such as “marginalizing over missing values” or “Pareto-smoothed importance sampling” could be briefly defined or paraphrased for clarity.

In summary, this is a promising and impactful study that requires clearer presentation and deeper contextualization. I encourage you to revise accordingly, and I look forward to seeing the improved version.

Reviewer #2: This manuscript meets the scientific, ethical, and reporting standards for publication in PLOS Mental Health however a light language edit focusing on sentence flow and stylistic consistency is needed.

**Do you want your identity to be public for this peer review?** For information about this choice, including consent withdrawal, please see our Privacy Policy

Reviewer #1: **Yes: ** Yalda Yazdani

Reviewer #2: No

---

## [Decision Letter · Decision Letter 1]

PMEN-D-25-00182R1

Functional improvement is a better predictor of steady work than medical improvement for individuals with mental health conditions

PLOS Mental Health

Dear Dr. Chang,

Thank you for submitting your manuscript to PLOS Mental Health. After careful consideration, we feel that it has merit but does not fully meet PLOS Mental Health’s publication criteria as it currently stands. Therefore, we invite you to submit a revised version of the manuscript that addresses the points raised during the review process.

We look forward to receiving your revised manuscript.

Kind regards,

David Onchonga, Ph.D.

Academic Editor

PLOS Mental Health

Journal Requirements:

Additional Editor Comments (if provided):

Reviewers' comments:

Reviewer's Responses to Questions

**Comments to the Author**

Reviewer #1: All comments have been addressed

Reviewer #2: All comments have been addressed

publication criteria?

Reviewer #1: Yes

Reviewer #2: Yes

3. Has the statistical analysis been performed appropriately and rigorously?

Reviewer #1: Yes

Reviewer #2: Yes

4. Have the authors made all data underlying the findings in their manuscript fully available (please refer to the Data Availability Statement at the start of the manuscript PDF file)?

Reviewer #1: Yes

Reviewer #2: Yes

5. Is the manuscript presented in an intelligible fashion and written in standard English?

Reviewer #1: Yes

Reviewer #2: Yes

Reviewer #1: The authors have done a very good job addressing previous comments, and the current manuscript is much improved. The study addresses a relevant and timely question, using a strong dataset and advanced analytical methods. The conclusions are well-supported, and the manuscript is generally clear and well-written.

At this stage, I have only a few minor suggestions for further improvement. First, in the introduction, the authors may consider adding one brief sentence to more explicitly highlight that direct comparisons between functional and medical improvements as predictors of employment are limited in the current literature — this would better frame the unique contribution of the study.

Second, while the methods are clearly described, a very brief statement explaining the rationale for using Bayesian methods (e.g., their flexibility with complex data and missing values) may help readers less familiar with these approaches. Similarly, a short sentence clarifying why the specific nonlinear models (piecewise regression and Bayesian neural networks) were selected would be helpful for completeness.

Third, although the discussion appropriately acknowledges the complex role of BMI, adding one or two lines further acknowledging possible reverse causality (e.g., weight changes as a result of functional decline) would strengthen the interpretation of these findings.

Lastly, the policy implications are well addressed; however, a brief addition about how these findings could inform disability determination processes in practice would increase the applied relevance.

Overall, these are small clarifications, and I believe the manuscript would be suitable for publication after minor revision.

Reviewer #2: The authors have addressed the comments raised in the earlier round of peer review. The manuscript now reflects improved clarity in structure, and detailed explanations of statistical procedures.

**Do you want your identity to be public for this peer review?** For information about this choice, including consent withdrawal, please see our Privacy Policy

Reviewer #1: **Yes: ** Yalda Yazdani

Reviewer #2: No

---

## [Editor Report · Decision Letter 2]

Functional improvement is a better predictor of steady work than medical improvement for individuals with mental health conditions

PMEN-D-25-00182R2

Dear Dr. Chang,

We are pleased to inform you that your manuscript 'Functional improvement is a better predictor of steady work than medical improvement for individuals with mental health conditions' has been provisionally accepted for publication in PLOS Mental Health.

Best regards,

David Onchonga, Ph.D.

Academic Editor

PLOS Mental Health